# Attitudes of the Public to Receiving Medical Care during Emergencies through Remote Physician–Patient Communications

**DOI:** 10.3390/ijerph17145236

**Published:** 2020-07-20

**Authors:** Matilda Hamlin, Steinn Steingrimsson, Itzhak Cohen, Victor Bero, Avishay Bar-Tl, Bruria Adini

**Affiliations:** 1Emergency Management & Disaster Medicine Department., School of Public Health, Sackler Faculty of Medicine, Tel-Aviv University, 39040 Tel Aviv, Israel; gushamlma@student.gu.se (M.H.); itzikco1@mail.tau.ac.il (I.C.); 2Institute of Neuroscience and Physiology, Sahlgrenska Academy, University of Gothenburg, 405 30 Gothenburg, Sweden; steinn.steingrimsson@vgregion.se; 3Meuhedet Health Services, Eben Gabirol 124, 62038 Tel Aviv, Israel; bero_victor@meuhedet.co.il (V.B.); avishay_bartal@meuhedet.co.il (A.B.-T.)

**Keywords:** remote communications, patient willingness, emergency management

## Abstract

Providing health services through remote communications for sub-acute health issues during emergencies may help reduce the burden of the health care system and increase availability of care. This study aimed to investigate the attitudes of the public towards receiving medical services and providing medical information through remote communication in times of emergencies. During the pandemic outbreak of the novel coronavirus (COVID-19), pandemic outbreak, 507 participants answered a structured online survey, rating their mean willingness to receive medical care and provide medical information, on a four-point Likert scale. Furthermore, demographic characteristics, social media use, and trust in data protection was collected. The mean willingness to receive medical services was 3.1 ± 0.6 and the mean willingness to provide medical information was 3.0 ± 0.7, with a strong significant correlation between the two (*r* = 0.76). The multiple regression model identified higher trust in data protection, level of education, and social media use as statistically significant predictors for a higher willingness to receive medical information while the first two predicted willingness to provide information. The findings suggest an overall positive attitude to receive medical care through remote communications.

## 1. Introduction

Major disasters and emergency events pose continual threats to the public health. The current ongoing pandemic outbreak of the novel coronavirus (COVID-19) constitutes a grave risk to the health of the global community and presents complex challenges in striving to contain the spread of the virus and mobilizing responses to maintain public health [1,2]. At the late end of December 2019, the first human cases of COVID-19 were observed in Wuhan, China [3,4,5]. The outbreak is currently ongoing, and as of the beginning of May 2020, over 3.4 million confirmed cases were identified globally [6].

To control a pandemic spread, strategies to minimize contact between infected and uninfected, must be utilized [1,7]. For example, in South Korea, during 2015, one single infected patient that visited a crowded emergency department (ED) caused 82 additional cases of the novel Middle East Respiratory Syndrome coronavirus (MERS-CoV), with a 20% attack rate of those who were in close contact with the patient [8]. Such strategies, to identify and isolate potentially infected and at the same time provide needed healthcare services, require cooperation between multiple organizations [1,7,9].

In Wuhan, triaging patients at a fever clinic, using a flow chart, determines the level of medical care. Infected patients with mild symptoms are referred to home isolation while only the severe cases are referred to the hospitals [10]. Isolation and home care are actions that must be taken to control the spread of a pandemic. However, in areas with widespread outbreaks, healthcare facilities find it challenging to offer proper isolation and conduct monitoring of all affected [9,10,11]. One effective way of monitoring and providing vital healthcare services to the isolated, without exposing additional individuals to the risk of infection, is through remote communication [12,13,14].

The definition of providing health services through remote communication, related to the definitions of telemedicine/telehealth/mHealth, is delivering various medical services using electronic tools. The interaction can be patient–clinician or clinician–clinician [15,16]. It is a rapidly developing field and an increasing component of all health services globally, where achievements in health status outcomes can be obtained [17,18]. Increased access, reduced travel time, less crowded facilities, and a reduced threshold for vulnerable groups are examples of how providing health services through remote communication can help optimize citizens’ health status [17,19].

In times of emergency, the vital resources of medical facilities, most especially acute-care hospitals, should primarily be dedicated to treating the moderate and severe patients, while looking for other alternatives to provide needed medical care for the less severe patients [20,21]. Providing health services through remote communications for patients with sub-acute illnesses during emergencies can reduce the burden of health care providers at these times [22,23,24]. 

Research shows that a video consultation as a replacement for an in-person triage in an ED is equal or superior to in-person patient safety and efficiency [25,26]. Kellerman et al. (2010) report how an online algorithm available to the public, used during the H1N1 Influenza Pandemic in 2009, helped reduce ED visits. Similar algorithms can be used to evaluate various diseases, not only to reduce ED visits but also to collect epidemiological information and characteristics of a new disease at an early stage [24]. Furthermore, health care workers are at a greater risk of being infected during pandemic outbreaks [27,28]. When asking health care workers at a local hospital about their attitudes towards working during a pandemic outbreak, almost 50% expressed resistance of going to work [28]. Remote communication can enable communication between the hospitals and the community and simultaneously decrease the exposure to pathogens for health care workers, as well as for patients with sub-acute illnesses. 

Remote communication may also be useful in stressful situations, such as natural catastrophes or human-made disasters, during which the public is apprehensive about mobility [29]. For example, an absence of effective communication between hospitals and the public was reported by Israeli experts in emergency management, referring to periods of conflicts in which the public was wary of leaving protected infrastructures. One expert expresses “In the 2006 war people sat in shelters and needed medical care and were afraid to go out to the health fund clinics” [30] (p. 4). Similarities can be drawn to providing medical care in rural areas, where there is often a lack of access to 24/7 medical care. Using remote communication for consultation and facilitating emergency care is incorporated in many of these rural areas [31,32,33].

There is a shortage of acute hospital beds in Israel with 1.8 per 1000 in the population and with an occupancy of 98% on average, far above the OECD average with 78% occupancy [34]. Given the ongoing pandemic COVID-19 combined with an absence of institutional reserve, there is a vital need to preserve acute hospital beds to severe and moderately ill patients and provide a well-organized monitoring of patients in home isolation through remote communications. 

This study aimed to investigate the attitudes of the public towards receiving medical services and providing medical information through remote communication during emergencies.

## 2. Materials and Methods 

### 2.1. Population

The study was conducted among a population insured by Meuhedet Health Services (MHS), which is one of four health funds operating in Israel, with over 1 million insured. This health fund was chosen for the study as it has advanced computerized systems for managing emergencies and was a leading actor in integrating telemedicine into the response plans for emergency scenarios. The sample size was calculated to 500 respondents, using the open source software for epidemiological statistics, OpenEpi at openepi.com. To encompass the two main populations that characterize the Israeli society, both Hebrew- and Arabic-speaking individuals were included. As the division of the two groups in the Israeli population is 4:1, 405 responses from the Hebrew-speaking population and 102 responses from the Arabic-speaking population were collected. Only individuals over 18 years of age were included in the study.

### 2.2. Procedures

A cross-sectional study was performed in the end of January through February 2020, in the midst of the COVID-19 outbreak, following the publication that several Israelis may have contracted the virus while visiting Wuhan, China. The study was planned by the researchers prior to COVID-19, resulting from the need to prepare for varied types of emergencies, including security scenarios and potential natural disasters. As COVID-19 erupted prior to the dissemination of the questionnaire to the respondents, it is not possible to disregard the impact of the pandemic on the answers of the participants. A Appendix A illustrates a timeline for the different study procedures (Appendix A).

### 2.3. Instrument 

The research tool used to collect data was a quantitative questionnaire developed specifically for this study, checked for content validity by 8 content experts and pilot tested among 20 participants, prior to its distribution. The questionnaire was conceptualized in Hebrew, and translated to Arabic (and re-translated to ensure validity) to enable both the majority and minority groups of Israel to partake in the study.

The first part of the questionnaire charted attitudes of the respondents concerning receiving medical services through remote communications during emergencies. Six statements focused on the attitudes towards receiving medical care from the health fund’s providers through remote communication (for example, receiving information about available medical facilities, instructions concerning needed medical treatment, distant medical examination through video-conference, etc.). Six other statements concentrated on attitudes towards providing the health fund with information or requesting medical/mental health services through remote communication (for example, updates concerning your current medical condition, transmit photos or video of your medical condition, request a home visit, etc.). A 4-point Likert scale was used ranging from 1 = “Not ready at all” to 4 = “Very ready” for each of the 12 different statements. 

The second part of the questionnaire included demographic questions on gender, age, place of residence, marital status, number of children under the age of 18/adult offspring living with the respondents, education, religion, religiosity, and socioeconomic status.

The third part of the questionnaire collected information on social media use, such as the frequency of activity and through which types of social media the respondents are willing to be contacted through in times of emergency. Furthermore, the third part contained a question concerning the trust of the data protection in transferring medical information through remote communication.

The questionnaire was distributed to members of MHS through an internet panel that employs a panel of over 100,000 individuals (http:www.ipanel.co.il). One of the exclusion criteria was members of other health funds, so that only members of this specific health fund were sampled. Further, through the panel, a representation of all demographic and geographic components of the population was conducted using stratified sampling methodology, based on quotas of age, gender, and geographic classification. The representation was based on data from the Central Bureau of Statistics.

### 2.4. Statistical Methods 

The Likert scale for statements 1–12 was converted into two different indices, one index for the willingness to receive medical services from a health fund through remote communications during emergencies and another index for the willingness to provide information to a health fund through remote communications during emergencies. The reliability of the indices was tested by Cronbach’s alpha and found to be very high (0.84 for willingness to receive medical services and 0.89 for willingness to provide information through remote communication).

To investigate differences in mean willingness between groups of characteristics, an independent samples t-test was calculated for dichotomous variables and an analysis of variance (one-way ANOVA) was calculated for variables with more than two values. When statistically significant values in the one-way ANOVA were presented, it was followed by Bonferroni’s post hoc test. The Pearson’s correlation coefficient (Pearson’s *r*) was calculated to examine a correlation between the willingness to receive medical services and the willingness to provide medical information through remote communication during emergencies. Spearman’s rank correlation coefficient (Spearman’s *r_s_*) was calculated to examine monotonic correlations between the willingness to either receive medical services or provide medical information and the ordinal variables social media use and trust of the data protection in transferring medical information through remote communication. The correlation was considered weak when coefficients *r* or *r_s_* ranged between −0.3 and 0.3; moderate between >−0.3 and −0.6 or >0.3 to 0.6; and strong between >−0.6 and −1.0 or >0.6 and 1.0. Finally, a prediction analysis was conducted using a multiple linear regression model. All statistical analysis was conducted using IBM SPSS Statistics version 26 (IBM Corp., Armonk, NY, USA). *p* value below 0.05 was considered significant. 

### 2.5. Ethical Aspects 

The study protocol was approved by ethical committees at the Tel Aviv University (# 0000734-2) and at Meuhedet Health Services (# P-01-29-01-20) according to local ethical guidelines. Each participant gave an informed consent after receiving information of the study before any data was collected. Furthermore, the study followed the Declaration of Helsinki principles in research ethics where participation was voluntary without a risk for breach of personal integrity. 

## 3. Results

### 3.1. Descriptive Analysis

The characteristics and descriptive statistics of the 507 respondents to the questionnaire are presented in Table 1. The mean age was 38.4 (standard deviation (SD) = 14.2) years and ranged between 18 and 70 years. The majority of the respondents had an income level near or above the average monthly salary (58.2%) and more than one child (54%). Self-reported frequency in use of social media was ≤1 a week (14.6%), several times a week (19.9%), and several times a day (65.5%). When rating the trust of data protection, 23.1% answered they “do not trust”, 36.5% answered they “trust”, and 40.4% “highly trust”. 

### 3.2. Differences in Mean Willingness to Receive and Provide Medical Services/Information

The mean willingness to receive medical services from the health fund through remote communications, in times of emergencies, was 3.1 (SD = 0.6). The mean willingness of the respondents to provide medical information to the health fund through remote communications, in times of emergencies, was 3.0 (SD = 0.7). Figure 1 shows the statistically significant differences that were found in mean willingness to receive medical services through remote communications during emergencies. According to age, the mean willingness to receive was significantly higher among those above 40 years of age compared to those under 40 years (*t*(505) = 2.1, *p* = 0.037). Similarly, the mean willingness to provide medical information through remote communications was found to be numerically higher in the age group above 40 years (mean 3.2, SD = 0.6 and mean 3.1, SD = 0.6, respectively); however, this was not statistically significant (*t*(505) = 1.91, *p* = 0.056). Figure 2 presents the statistically significant differences in the mean willingness to provide medical information through remote communications during emergencies. 

Socioeconomic-related differences were statistically significant in the mean willingness to both receive medical services (*F*(2504) = 3.48, *p* = 0.032) and provide medical information through remote communication (*F*(2504) = 3.14, *p* = 0.044). Post hoc analysis using Bonferroni’s multiple comparison showed higher levels of mean willingness to receive medical services among respondents with an above-average salary compared to respondents with a salary near average (*p* = 0.049); however, the multiple comparison was not statistically significant in willingness to provide. 

Significant differences were seen between the respondents according to their level of trust of the data protection when transferring medical information through remote communications during emergencies, in both mean willingness to provide medical information (*F*(2504) = 54.36, *p* < 0.001) and to receive medical services (*F*(2504) = 44.11, *p* < 0.001). Further post hoc analysis, using the Bonferroni multiple comparison, showed that the mean willingness of both providing and receiving was higher between all groups that reported higher levels of trust of data protection (*p* < 0.001). There were no significant differences between different religions, which is equivalent to the two language categories, Hebrew and Arabic. Furthermore, gender, living area, religiosity, marital status, education, and social media use were not associated with a higher mean willingness to receive or provide medical care/information. 

### 3.3. Correlations between Willingness to Receive or Provide Medical Services/Information and Other Variables

There was a significantly strong positive linear correlation between willingness to provide medical information and willingness to receive medical services through remote communication during emergencies (*r* = 0.763, *p* < 0.001). 

Significant moderate positive monotonic correlations were found between trust in data protection in transferring medical information through remote communication and both the willingness to provide (*r_s_* = 0.43, *p* < 0.001) as well as the willingness to receive (*r_s_* = 0.38, *p* < 0.001), i.e., higher willingness was associated with higher trust. 

### 3.4. Linear Regression Model

A linear regression model was conducted in order to examine which variables can predict the willingness of the respondent to receive medical services through remote communications. The ANOVA test was significant (*F*(3368) = 20.02, *p* < 0.001) and showed a good fit of the model (R^2^ = 0.14). This means that in total, 14% of the variance of a respondent’s willingness to receive medical services through remote communications can be explained by using three variables: The respondent’s level of trust in data protection (*p* < 0.001), respondent’s level of education (*p* = 0.038), and respondent’s level of activity on social media (*p* = 0.043). Among these three variables, the trust of data protection has the highest effect on respondents’ willingness to receive medical services (B-coefficient = 0.347). The linear regression model is presented in Table 2. Gender, age, marital status, religion, religiosity, socio-economic status, number of children under the age of 18, or other dependent adults living with the respondents were not statistically significant in the multiple analysis and therefore were not included in the model.

Similarly, a linear regression model was conducted to examine the predicting variables in the willingness to provide medical information through remote communications. The ANOVA test was significant (*F*(2369) = 35.8, *p* < 0.001), with R^2^ = 0.16. In total, 16% of the variance of a respondent’s willingness to provide medical information through remote communications can be explained by using two variables: The respondent’s level of trust in data protection (*p* < 0.001) and the level of education (*p* = 0.015). Among the two variables, the trust of data protection has the highest effect on respondents’ willingness to provide medical information (B-coefficient = 0.389). The linear regression model is presented in Table 3. Gender, age, marital status, religion, religiosity, socioeconomic status, number of children under the age of 18, or other dependent adults living with the respondents and level activity on social media were not statistically significant in the multiple analysis and therefore were not included in the model. 

## 4. Discussion

This study, examining attitudes to receive health services through remote communications in times of emergency, was conducted during an emergency situation—the COVID-19 crisis. Though the actual cases that were infected in the Israeli communities were only confirmed at a later date, the study was conducted during a period in which the population was notified of an evolving epidemic that though (at that time) it impacted other countries, it was expected to also impact the local society. Therefore, it can be assumed that the results of this study realistically reflect the attitudes and perceptions of the public concerning consumption of services through remote communication, in a way which could not have been achieved otherwise [35]. In this situation, when mobility in society is restricted and medical facilities are primarily dedicated to treating the moderate and severe patients, it is more likely that respondents can relate and prefer health services through remote communications as an alternative way to receive needed medical care for sub-acute illnesses and monitoring of chronic illnesses [36]. Furthermore, it may reflect their concern from being exposed to the risk, such as contracting the COVID-19 if they were to physically go to the medical facilities [37], and thus consent to utilizing the services through telemedical means.

The willingness of the general public to provide medical information and to receive medical services from a health fund through remote communications in times of emergency was found to be high. By testing each factor’s effect on patient’s willingness to receive medical services and provide medical information, the factors associated with a higher willingness to receive medical services were age, salary in relation to average income, and trust of the data protection while for willingness to provide medical information, the latter two were statistically significant. The respondents scored very similar in willingness to receive medical services and provide medical information to a health fund through remote communications in times of emergency. Not surprisingly, there was a significant strong positive correlation between the two variables. The predictors of willingness to receive medical services by using a multiple model were trust of data protection transferring medical information through remote communications, level of education, and level of activity on social media. Similarly, the predictors of willingness to provide medical information were trust of data protection and level of education. The R^2^ value represents the percent of the variance of the dependent variable that can be explained by using the regression model including the independent variables. The prediction values were relatively low, which could be explained as there was a low variance in the willingness reported by the participants. Further, including other independent variables in order to predict the dependent variables could have increased the R^2^ value.

According to the conducted literature review, no previous studies concerning attitudes towards health services through remote communications in times of emergency were identified. However, the willingness to use telemedical means among patients with specific medical conditions, such as chronic diseases with increased risk of cardiovascular disease or depression, has been studied [38,39,40]. In one of them, an American study investigating attitude towards telemedical means when monitoring diabetic retinopathy, Valikodath et al. reported (2017) that 97% of their respondents had never been in contact with the word telemedicine before [39]. This indicates that there are barriers to overcome when implementing telemedical means in general; however, patient satisfaction rates are reported as very high in the majority of studies when it is implemented [41,42,43]. 

The trust of data protection transferring medical information through remote communications is the one factor that remained statistically significantly associated with willingness to use remote communications in all our analyses. These results are in accordance with previous studies [44,45,46]. In a Dutch study, trust of the data protection in transferring medical information through remote communications is shown to be the one sub-factor that affects the overall trust in using telemedicine the most, with a higher impact than both trust in health care providers, health care workers, and offered treatment [44]. Furthermore, a German study showed how the trust of data protection is of great importance, especially among healthy respondents, when considering the use of health services through remote communications [46].

This study indicates that socioeconomic differences might affect the attitude towards medical services through remote communications. An American study [47] found that the level of education was associated with increased use of health services through remote communications. Similarly, Green and colleagues found that a lower education level was associated with a higher likelihood in refusing to participate in a trial to monitor and optimize blood pressure through remote communications [48]. 

Level of activity on social media was a predictor in the willingness to receive medical services through remote communications. In a recent study, it was shown that in an ED-visiting population with a low socioeconomic status, 96% had access through a smart phone [40], indicating that in high-income regions, the access to social media is high regardless of socioeconomic status.

Although age was not found to be an independent predictor, it is surprising that younger age was not a strong predictor. Even though technological use is increasing in the older generation [49], it is common that the younger generation uses technology to a greater extent. It is reasonable to believe that more frequent use and convenience in technology would correlate with a higher willingness towards telemedicine. In accordance with this consideration, a Polish study examining the willingness to use telemedicine among those 60 years and older showed that 66% of the respondents were reluctant to conduct a video consultation with a physician [50]. One possible explanation to the counterintuitive age-related results found in our study can be that in times of emergency, the older generation are more likely to trust the authorities in emergency management compared to the younger generation. It may also reflect a higher degree of concern with health-related issues among older individuals compared to younger people, who tend to be generally healthier. More so, as the elderly were defined as highly vulnerable to COVID-19, their willingness to access services through remote communication may signify their perceived concern of leaving their homes and thus a higher preference for maintaining their safety by not being potentially exposed to the virus. The younger populations, perceiving themselves as less vulnerable, were most probably less concerned with the need to strictly maintain social distancing [51].

### Limitations

The cross-sectional design cannot determine causality nor capture changes in attitudes over time. Given this, it is difficult to assess the extent of the impact that COVID-19 has had on the results. Future research is needed to evaluate the differences in usage and patient satisfaction of telemedical means, actually implemented in clinical practice during the COVID-19 pandemic outbreak.

Furthermore, a strength of the study is how the internet panel, with over 100,000 individuals, made it possible to stratify the sampling, based on quotas of age, gender, and geographic classification, enabling a comparison to the Israeli population and also to collect the exact calculated sample size. However, distributing the questionnaire exclusively through an internet panel when the aim was to investigate willingness to access health services using electronic tools could exclude groups in the society who do not have access or are reluctant to use electronic devices. If the questionnaire were to be conducted also by post or phone-call, respondents with less access to electronic tools may have been more fully represented.

## 5. Conclusions

In this cross-sectional study, conducted during an ongoing worldwide pandemic outbreak, the willingness to receive medical services and to provide medical information through remote communications in times of emergency was found to be high. There are factors associated with statistically significant differences in the attitude towards health services through remote communications, where the trust of data protection in transferring medical information through remote communications was the factor with the highest impact. The results are important in future implementation of these solutions in clinical practice.

## Figures and Tables

**Figure 1 ijerph-17-05236-f001:**
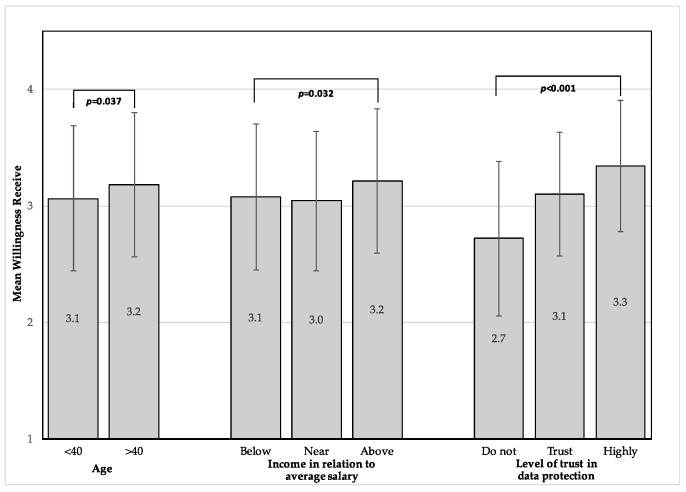
Differences in mean willingness to receive medical services through remote communications in times of emergencies. Error bars represent standard deviation.

**Figure 2 ijerph-17-05236-f002:**
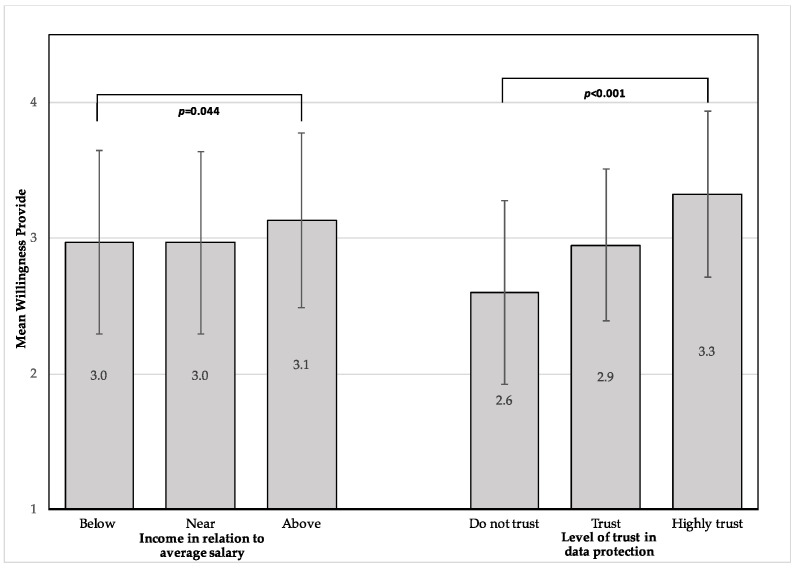
Differences in mean willingness to provide medical information through remote communications in times of emergencies. Error bars represent standard deviation.

**Table 1 ijerph-17-05236-t001:** Characteristics of respondents to questionnaire (*n* = 507).

Characteristics		*n* (%)
Age	<40	299 (59.0)
Gender	Male	274 (54.0)
Living area	Jerusalem	71 (14.0)
	Tel Aviv-Gush Dan-Sharon	185 (36.5)
	Haifa and North	182 (35.9)
	South and Lowlands	69 (13.6)
Religion	Jewish	410 (80.9)
	Non-Jewish	97 (19.1)
Religiosity	Secular	256 (50.5)
	Traditional	151 (29.8)
	Religious	100 (19.7)
Marital status	In partnership	347 (68.4)
Children <18 living at home	No children	232 (45.8)
	One child	90 (17.8)
	≥2 children	185 (36.5)
Adult offspring at home	No children	233 (46.0)
	One child	109 (21.5)
	≥2 children	165 (32.5)
Education	Below tertiary education	247 (48.7)
	Tertiary education	260 (51.3)
Salary	Below average	212 (41.8)
	Near average	133 (26.2)
	Above average	162 (32.0)

**Table 2 ijerph-17-05236-t002:** Linear regression model: willingness to receive medical services and other variables.

Variable	Unstandardized Coefficients		Standardized Coefficient		
	B	Standard Error	B	t	*p*
(Constant)	2.324	0.126		18.462	<0.001
Trust of data protection	0.263	0.037	0.347	7.162	<0.001
Education	0.119	0.057	0.100	2.077	0.038
Activity on social media	0.076	0.037	0.098	2.032	0.043

**Table 3 ijerph-17-05236-t003:** Linear regression model: willingness to provide medical information and other variables.

Variable	Unstandardized Coefficients		Standardized Coefficient		
	B	Standard Error	B	t	*p*
(Constant)	2.273	0.095		23.899	<0.001
Trust of data protection	0.319	0.039	0.389	8.161	<0.001
Education	0.149	0.061	0.117	2.452	0.015

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
