# Peer review of "Attitudes of the Public to Receiving Medical Care during Emergencies through Remote Physician–Patient Communications"

_ijerph, 2020, doi:10.3390/ijerph17145236_

Round 1
Reviewer 1 Report
Thank you for the opportunity to review this paper on the influence of COVID-19 on the physician-patient relationship.
An internet-based survey was undertaken in Israel in January of 2020. In several places in the paper, the authors refer to the possible influence and effect of the Pandemic on the responses received. The reality is that there had been no cases of COVID-19 infection in Israel by January 2020. The WHO declared the Pandemic on 11 March, and according to https://www.worldometers.info/coronavirus/ the first reported case of COVID-19 infection in Israel was towards the end of February and the first death in mid to late March. The total reported deaths worldwide by the end of January was fewer than 300.
Examples of misleading statements are:
Lines 226-227: “This study, examining attitudes to receive health services through remote communications in times of emergency, was conducted during an emergency situation – the COVID-19 crisis.”
Lines 234-236: “Furthermore, it may reflect their concern from being exposed to the risk, such as contracting the COVID-19 if they were to physically go to the medical facilities [35], and thus consent to utilizing the services through telemedical means.”
Lines 284-292: “One possible explanation to the counter-intuitive age-related results found in our study can be, that in times of emergency, the older generation are more likely to trust the authorities in emergency management compared to the younger generation. More so, as the elderly were defined as highly vulnerable to COVID-19, their willingness to access services through remote communication may signify their perceived concern of leaving their homes and thus a higher preference to maintaining their safety by not being potentially exposed to the virus. The younger populations, perceiving themselves as less vulnerable were most probably less concerned with the need to strictly maintain social distancing [49].
Lines 303-305: “In this cross-sectional study, conducted during an ongoing worldwide pandemic outbreak, the willingness to receive medical services and to provide medical information through remote communications in times of emergency was found to be high”.
Lines 249-250: “According to the conducted literature review, no previous studies concerning attitudes towards health services through remote communications in times of emergency were identified.”
88-89: “This study aims to investigate the attitudes of the public towards receiving medical services and providing medical information through remote communication during emergencies.” This may have been the aim of the study, but the study must surely then have been conceived before the emergency and Pandemic.
The methods are inconsistent and confusing. In line 93, it is stated that “The study was conducted among a population insured by Meuhedet Health Services” which insures over 1 million people. Lines 123-126 make no mention of participants being insured by this health service, “The questionnaire was distributed using an internet panel that employs a panel of over 100,000 individuals, representing all demographic and geographic components of the population (http:www.ipanel.co.il). Stratified sampling methodology was used, based on quotas of age, gender and geographic classification, based on the Central Bureau of Statistics.” Nor do Lines 293-295, “A strength of the study is how the internet panel with over 100,000 individuals made it possible to stratify the sampling, comparable to the Israeli population, and also to collect the exact calculated sample size making it applicable on the Israeli population.” Which version is correct?
What was the rationale for choosing the Meuhedet Health Service?
What is the relationship of the authors with this service?
The authors do not take into account the relevant literature on acceptance of telemedicine during natural disasters which are also emergencies.
There is no mention of ethical approval for the study or participants consenting to take part.
Line 167: data are missing, “…numerically higher in the age group above 40 years (mean xx, SD = xx and mean xx, SD = xx…”
Table 1: what is meant by “academic”? Presumably, it refers to tertiary/university education.
Figures 1 and 2: why has standard error of the mean been used in the figures when standard deviation is reported every else in the paper.
When data are collected as whole numbers, as in the Likert scale used, the mean should be reported to one decimal place at the most.
The paper needs minor English editing.
The data reported are interesting. The validity of their contextualisation is questionable.
Author Response
Thank you for your review of our manuscript. Below is each comment numbered followed by an answer. The changes made are referred in the answer and changes highlighted in the revised manuscripts.
Comment 1: An internet-based survey was undertaken in Israel in January of 2020. In several places in the paper, the authors refer to the possible influence and effect of the Pandemic on the responses received. The reality is that there had been no cases of COVID-19 infection in Israel by January 2020. The WHO declared the Pandemic on 11 March, and according to https://www.worldometers.info/coronavirus/ the first reported case of COVID-19 infection in Israel was towards the end of February and the first death in mid to late March. The total reported deaths worldwide by the end of January was fewer than 300. Examples of misleading statements are: Lines 226-227: “This study, examining attitudes to receive health services through remote communications in times of emergency, was conducted during an emergency situation – the COVID-19 crisis.” / Lines 234-236: “Furthermore, it may reflect their concern from being exposed to the risk, such as contracting the COVID-19 if they were to physically go to the medical facilities [35], and thus consent to utilizing the services through telemedical means.” / Lines 284-292: “One possible explanation to the counter-intuitive age-related results found in our study can be, that in times of emergency, the older generation are more likely to trust the authorities in emergency management compared to the younger generation. More so, as the elderly were defined as highly vulnerable to COVID-19, their willingness to access services through remote communication may signify their perceived concern of leaving their homes and thus a higher preference to maintaining their safety by not being potentially exposed to the virus. The younger populations, perceiving themselves as less vulnerable were most probably less concerned with the need to strictly maintain social distancing [49]. / Lines 303-305: “In this cross-sectional study, conducted during an ongoing worldwide pandemic outbreak, the willingness to receive medical services and to provide medical information through remote communications in times of emergency was found to be high”. / Lines 249-250: “According to the conducted literature review, no previous studies concerning attitudes towards health services through remote communications in times of emergency were identified.” / 88-89: “This study aims to investigate the attitudes of the public towards receiving medical services and providing medical information through remote communication during emergencies.” This may have been the aim of the study, but the study must surely then have been conceived before the emergency and Pandemic.
Answer 1: This is a very important comment which we understand may cause confusion to the reader as the reviewer correctly points out. The study was conducted from the end of January until mid- February; during this period, although the epidemic had not as yet been declared by the WHO as a global pandemic, the Israeli authorities as well as the press notified the public that there was an evolving epidemic that would be expected to impact the local population, furthermore, it was reported that several Israelis might have contracted the virus while visiting Wuhan, China. The risk perception of the Israeli population was therefore highly elevated, which elicited grave concerns among the varied groups of the public. To ensure that the readers of the article understand this we have added further clarification to the article. Please see page 2-3 lines 93-99 and page 7, lines 244-247.
Comment 2: The methods are inconsistent and confusing. In line 93, it is stated that “The study was conducted among a population insured by Meuhedet Health Services” which insures over 1 million people. Lines 123-126 make no mention of participants being insured by this health service, “The questionnaire was distributed using an internet panel that employs a panel of over 100,000 individuals, representing all demographic and geographic components of the population (http:www.ipanel.co.il). Stratified sampling methodology was used, based on quotas of age, gender and geographic classification, based on the Central Bureau of Statistics.” Nor do Lines 293-295, “A strength of the study is how the internet panel with over 100,000 individuals made it possible to stratify the sampling, comparable to the Israeli population, and also to collect the exact calculated sample size making it applicable on the Israeli population.” Which version is correct?
Answer 2: Thank you for pointing out this methodological inconsistency and confusion. To answer the question, the internet panel possessed information about which Health Fund the respondents belong to, and those who belonged to other Health Funds were excluded from the study. Therefore, it was possible to exclusively ask respondents who are insured by Meuhedet Health Services through the internet panel to participate and at the same time use stratified sampling methodology. To clarify this confusion, changes have been made at page 3, lines 128-134 and page 8, lines 317-319.
Comment 3: What was the rationale for choosing the Meuhedet Health Service?
Answer 3: This Health Fund was chosen for the study as it has advanced computerized systems for managing emergencies and was a leading entity in initiating telehealth as part of the response plans for emergency scenarios. We have added this clarification to the article. Please see page 3, lines 95-99.
Comment 4: What is the relationship of the authors with this service?
Answer 4: Two of the authors belong to the emergency unit within the Health Fund and as such perceived that it is of utmost importance to examine whether telehealth can be effectively used during times of emergencies, and thus approached the academic colleagues with a request to conduct this study. Furthermore, a concurrent study was conducted among the personnel of the Health Fund (they’re the other important chain in this initiative) to identify their willingness to use telehealth to communicate with patients. This will be reported in another article (it was too much information to include in one article).
Comment 5: The authors do not take into account the relevant literature on acceptance of telemedicine during natural disasters which are also emergencies.
Answer 5: The comment is very reasonable, acceptance of telemedicine during natural disasters is relevant to the field. In accordance with this, additional text to the introduction has been added on page 2, lines 75-76 and a reference has been added.
Comment 6: There is no mention of ethical approval for the study or participants consenting to take part.
Answer 6: The observation is correct. The ethical approval of the study is now mentioned at page 4, lines 157-162.
Comment 7: Line 167: data are missing, “…numerically higher in the age group above 40 years (mean xx, SD = xx and mean xx, SD = xx…”
Answer 7: Thank you for discovering and mentioning our omission. The missing data has been added to page 5, line 182.
Comment 8: Table 1: what is meant by “academic”? Presumably, it refers to tertiary/university education.
Answer 8: As mentioned in the comment by the reviewer, “academic” refers to tertiary/university education. However, we agree that tertiary/university education is a better way of expressing the level of education. Changes have been made in the manuscript so academic is referred to as tertiary education and non-academic is referred to as below tertiary education. Please see page 4, Table 1.
Comment 9: Figures 1 and 2: why has standard error of the mean been used in the figures when standard deviation is reported every else in the paper.
Answer 9: Thank you, it is indeed the standard deviation value which should be reported throughout the article. Please see page 5, figure 1 and page 6, figure 2.
Comment 10: When data are collected as whole numbers, as in the Likert scale used, the mean should be reported to one decimal place at the most.
Answer 10: Thank you for a comment which we agree upon. All mean values of the Likert scale mentioned in the article have been changed to one decimal only.
Comment 11: The paper needs minor English editing.
Answer 11: We have read the manuscript and tried to adjust our language in accordance with the journal´s standard.
Comment 12: The data reported are interesting. The validity of their contextualisation is questionable.
Answer 12: We thank the reviewer for acknowledging the data and hope we have clarified the contextualization.
Reviewer 2 Report
This paper is nicely done. I have only a few comments:
* Cites are needed for the sentence on page 2, lines 55-56.
* The paragraph on page 2, lines 78 to 82, is problematic to me because I do not see clear links between the factors mentioned and the research question.
* It would have been useful for the survey instrument to have undergone cognitive testing, which is in my experience far superior to "pre-testing" in determining whether an instrument is understood as intended by the study population. In particular, I would have wanted to see comparisons between Jews and Arabs in how they understood items.
* I am a little uncertain about whether survey respondents had any relationship to the Health Plan with which some authors were connected. What if anything was the role of the Health Plan in the study? Was everyone in the study in one Health Plan or were they affiliated with different ones? Was anyone without a link to a plan?
* I find the Rsquared values, those statistically significant, to be pretty low. Can you comment on that? What might it mean?
* On page 7, lines 251-252, you say that willingness use telemedical means in other medical fields had differed. Can you specify the fields and why people's responses might be different?
* On page 8, you note surprise that older rather than younger people were more open to telemedicine. I think this reflects a higher degree of concern with health among older people. It is a shame you did not include a self-reported health question as that information would have been relevant and useful.
Author Response
Thank you for your review of our manuscript. Below is each comment numbered followed by an answer. The changes made are referred in the answer and changes highlighted in the revised manuscripts.
Comment 1: Cites are needed for the sentence on page 2, lines 55-56.
Answer 1: Thank you for observing the need of references to the sentence, which are now added in page 2, line 56.
Comment 2: The paragraph on page 2, lines 78 to 82, is problematic to me because I do not see clear links between the factors mentioned and the research question.
Answer 2: Thank you for highlighting the absence of a clear link between the mentioned paragraph and the research question. To clarify the linkage, the paragraph has been transferred to page 2, lines 69-74 and additional text has been added.
Comment 3: It would have been useful for the survey instrument to have undergone cognitive testing, which is in my experience far superior to "pre-testing" in determining whether an instrument is understood as intended by the study population. In particular, I would have wanted to see comparisons between Jews and Arabs in how they understood items.
Answer 3: The reviewer has a very good point which we agree upon and will consider in future research. Unfortunately, no cognitive testing was conducting in this study. However, we added an additional sentence in the result section at page 5, lines 197-199 to clarify that there were no significant differences in willingness between the different language categories.
Comment 4: I am a little uncertain about whether survey respondents had any relationship to the Health Plan with which some authors were connected. What if anything was the role of the Health Plan in the study? Was everyone in the study in one Health Plan or were they affiliated with different ones? Was anyone without a link to a plan?
Answer 4: All of the survey respondents were insured by this specific Health Plan; all those that were insured by one of the other Health Funds were excluded from the study (all residents in Israel have to belong to one of four Health Funds). This has been added to the article. See page 3 lines 128-134.
Two of the authors belong to the emergency unit within the Health Fund and as such perceived that it is of utmost importance to examine whether telehealth can be effectively used during times of emergencies, and thus approached the academic colleagues with a request to conduct this study. Furthermore, a concurrent study was conducted among the personnel of the Health Fund (they’re the other important chain in this initiative) to identify their willingness to use telehealth to communicate with patients. This will be reported in another article (it was too much information to include in one article).
This Health Fund was chosen for the study as it has advanced computerized systems for managing emergencies and was a leading entity in initiating telehealth as part of the response plans for emergency scenarios. We added this clarification to the article. Please see page 2-3, lines 93-99.
Comment 5: I find the Rsquared values, those statistically significant, to be pretty low. Can you comment on that? What might it mean?
Answer 5: Thank you for pointing out this important observation. The R2 value represents the percent of the variance of the dependent variable which can be explained by using the regression model including the independent variables. The prediction values were relatively low, which could be explained by that there was not a big variance in the willingness variable. Further, including other independent variables in order to predict the dependent variables could have increase the R2 value. We have added a comment concerning this in the discussion page 7-8, lines 269-273.
Comment 6: On page 7, lines 251-252, you say that willingness to use telemedical means in other medical fields had differed. Can you specify the fields and why people's responses might be different?
Answer 6: We agree that this section can be clarified, therefore, an addition has been made on page 8, Lines 275-277.
Comment 7: On page 8, you note surprise that older rather than younger people were more open to telemedicine. I think this reflects a higher degree of concern with health among older people. It is a shame you did not include a self-reported health question as that information would have been relevant and useful.
Answer 7: We find the comment of the reviewer very important and relevant. We thus added (as the reviewer suggested) that this finding may also reflect a higher degree of concern with health-related issues among older individuals compared to younger people, who tend to be generally healthier. Please see page 8, lines 310-312. As for the lack of a self-reported health question, we noted this and will implement it in future relevant studies.
Round 2
Reviewer 1 Report
Thank you for the opportunity to review this paper again.
A previous comment was about the study being undertaken in January 2020 as the first COVID-19 case in Israel was confirmed on 21 February. The previous version of the paper reported, “A cross-sectional study was performed in January 2020 in the midst of the COVID-19 outbreak.” This has been changed to “was performed in the end of January through February in the midst of the COVID-19 outbreak…” While this addresses the concern raised, it brings the veracity of the paper as a whole into question.
Author Response
We thank the reviewer for the comment. We have made extensive changes based on the academic editors’ comments.
The initial statement of the dates in which the study was conducted was made based only on the date of the ethics approval (January 29th 2020). As the reviewer and academic editor (most correctly) pointed to the need to state the exact dates, we went back to the data and stated the accurate dates of each activity of the study included under a new subheading and illustrated in a figure (lines 103-113). Furthermore, we have changed the title in order to minimize risk for confusion.